# Preparation and Characterization of Novel Multifunctional Wound Dressing by Near-Field Direct-Writing Electrospinning and Its Application

**DOI:** 10.3390/polym16111573

**Published:** 2024-06-01

**Authors:** Dingfan Li, Dongsong Lin, Yun Li, Sikun Xu, Qingyun Cao, Wuyi Zhou

**Affiliations:** 1Biomass 3D Printing Research Center, College of Materials and Energy, South China Agricultural University, Guangzhou 510642, China; lin039@foxmail.com (D.L.); 20222158016@stu.scau.edu.cn (D.L.);; 2Guangdong Yunzhao Medical Technology Co., Ltd., Guangzhou 510000, China; 3College of Animal Science, South China Agricultural University, Guangzhou 510642, China

**Keywords:** near-field electrospinning, polycaprolactone, collagen, erythromycin

## Abstract

Near-field direct-writing electrospinning technology can be used to produce ordered micro/nanofiber membrane dressings. The application of this technology can simply realize the control of dressing porosity, compound different functional substances, and adjust their distribution, thus improving the defects of common dressings such as insufficient breathability, poor moisture retention performance, and single function. Herein, a novel multifunctional wound dressing was prepared to utilize near-field direct-writing electrospinning technology, in which calf skin collagen type I (CSC-I) and polycaprolactone (PCL) were used as the composite matrix, Hexafluoroisopropanol (HFIP) as the solvent, and erythromycin (ERY) as an anti-infective drug component. The results show that the micro/nanofiber membranes prepared by near-field direct-writing electrospinning technology can all present a complete mesh structure, excellent thermal stability, and good moisturizing properties. Moreover, the composite fiber membrane loaded with ERY not only had obvious antibacterial properties against *E. coli* and *S. thermophilus* but also a better slow-release function of drugs (it is rare to have both in traditional wound dressings). Therefore, this experimental design can provide relevant theories and an experimental foundation for preparing a new type of medical dressing with drug loading and has good guiding significance for the application and promotion of near-field direct-writing electrospinning in medical dressings.

## 1. Introduction

Skin is the outermost layer of the human body and also plays a role as the first line of defense for the human immune system. Skin can help protect marcels, ligaments, and other inner organs from the influence of external environmental factors such as UV rays, oxidation, dehydration, and pathogenic microorganisms [1,2,3]. To stop bleeding faster and relieve a patient’s pain after the skin is injured, some researchers have focused on developing wound dressings [4,5,6].

It is believed that wound dressings should be economical, easy to use, and readily available. The dressings or method of coverage will provide good pain relief as well as protection from infection, promotion of healing, moisture retention, elasticity, antigen-free properties, and good adhesion. Conventional wound dressings, represented by gauze and sutures, can temporarily cover wounds to stop bleeding and protect the wound from bacterial infection. Those kinds of wound dressings show excellent softness, affinity to skin, biodegradability, and regeneration properties as well [7,8]. However, they have poor biocompatibility, moisture retention ability, and single function. For example, conventional wound dressings made of cotton or synthetic fibers can help keep wounds dry, but studies have shown that dry healing is detrimental to epithelial tissue growth [9]. In addition, during chronic wound healing, conventional wound dressings are difficult to ameliorate bacterial-induced wound deterioration and are also prone to mechanical re-injury during dressing changes [10]. When conventional wound dressings are used in conjunction with drugs, it is difficult to achieve sustained drug release; rapid drug release tends to result in more pronounced stimulation. As a result, patients may suffer more when using them.

Hydrogel, as one of the rising dressings, is composed of hydrophilic polymer chains and has adjustable physical and chemical properties. Its three-dimensional (3D) network structure, similar to that of macromolecules in the human body, is superior to other wound dressings in terms of biocompatibility, maneuverability, and non-invasiveness. In addition, it is exceptionally unique in its ability to deliver drugs and genes [11,12,13]. Collagen provides strength, integrity, and structure to normal tissues, which helps with wound healing [12,13]. In recent decades, collagen hydrogels have begun to be known and researched for superiorities like their ability to contain functional drugs to achieve sustained drug release. However, collagen hydrogels still have defects such as low mechanical properties, breathability, tissue adhesion, and insufficient stability, which require other carriers for support [14,15,16].

Materials produced by near-field direct-writing electrospinning technology have the advantages of a large specific surface area, good mechanical properties, and surface characteristics, so they can be widely used in specific fields such as conductive fibers, biomedical polymer materials, and lightweight composite materials [17,18,19]. The emerging near-field direct-writing electrospinning technology has the characteristics of customizable material shape, size, diameter, and properties of polymeric nanofibers [20,21]. In addition, polymeric nanofibers possess a high surface-to-volume ratio that could enhance cell attachment and promote wound healing [22,23]. If combined with medical dressings, it can customize the drug distribution and topological structure of dressings based on the patient’s wound characteristics and structure, overcome the shortcomings of the insufficient breathability of new dressings, and have the potential for tissue engineering applications [24]. 

Polycaprolactone (PCL) has a low melting temperature, a low glass-transition temperature, high thermal stability, excellent processability, excellent mechanical properties, biocompatibility, and biodegradability, and is highly permeable to a variety of drugs and easy to load with drugs, which has stimulated researchers [25,26,27]. Extensive research has been conducted on its potential applications in the biomedical field [28]. 

Collagens, which are the main structural proteins responsible for the structural integrity of vertebrates and many other multicellular organisms, contain three proteins wrapped around each other in a triple-helix structure. Collagen makes up a large portion of the extracellular matrix, acts as a structural scaffold in tissues, and affects cellular functions such as differentiation, migration, and the synthesis of proteins [29,30]. When collagen is added to dressings, it makes dressings perform better in terms of convenient application, naturalness, non-immunogenicity, non-pyrogenicity, hypoallergenicity, and pain-free properties [31]. Calf skin collagen type I (CSC-I), as a typical collagen, has good hemostatic properties, biocompatibility, low immunogenicity, and biodegradability [32]. It is beneficial to promote the growth and repair of skin [33], retain water in the dermis, and maintain its hydration and gloss [34]. Erythromycin (ERY), whose molecular formula is C_37_H_67_NO_13_, can act on ribosomes through the bacterial cell membrane. There are two ways of action: on the one hand, it inhibits the formation of the 50S ribosome large subunit; on the other hand, it inhibits the translation of ribosomes, thereby achieving the purpose of inhibiting bacterial reproduction [35,36,37]. The use of ERY during the wound healing process can prevent and treat wound infections caused by Gram-positive bacteria and Gram-negative bacteria.

Herein, Hexafluoroisopropanol (HFIP) was used as the solvent, PCL and CSC-I were used as the composite substrate, ERY was added as the active drug, and a new type of fiber with regular shape, uniform diameter, and good air permeability was spun using near-field electrospinning technology. As a medical dressing, this dressing has good mechanical properties and thermal stability and has obvious antibacterial, slow-release, and moisturizing functions. This work not only proposes a facile method for the fabrication of multifunctional new wound dressings but also has great significance for the development of new wound dressings and the expansion of near-field electrospinning applications.

## 2. Materials and Methods

### 2.1. Materials Required

Calf skin collagen type I (CSC-I, Mn = 3000 g/mol) was bought from Xi’an Yunhe Biotechnology Co., Ltd. (Xi’an, China). Hexafluoroisopropanol (HFIP), polycaprolactone (PCL), and methyl green were purchased from Shanghai McLean Biochemical Technology Co., Ltd. (Shanghai, China). Phosphate buffer saline (PBS, pH = 7.2~7.4, 0.01 M) was purchased from Jinclone (Beijing) Biotechnology Co., Ltd. (Beijing, China). Beef extract and peptone for bacteriology were acquired from Guangdong Huankai Microbiology Technology Co., Ltd. (Guangzhou, China).

### 2.2. Experimental Methods

#### Preparation of Composite Fiber Membrane

Firstly, 18 wt% PCL solution in HFIP solvent was prepared; then, 0, 10, 20, and 30% CSC-I (x% content in this experiment represents x% of the PCL mass) was added, and the mix was stirred, dissolved in a 50 °C water bath, and let stand. After defoaming, CSC-I/PCL spinning solutions with different CSC-I contents were prepared. Due to 30% CSC-I being susceptible to needle clogging during the spinning process, ERY/CSC-I/PCL spinning liquids with different ERY contents were prepared by taking 20% CSC-I and adding 1, 3, and 5% ERY (x% content represents x% of the PCL mass) under the same conditions as above.

A near-field direct-writing electrospinning machine (NFES01-001, Foshan Lepton Precision Measurement and Control Technology Co., Ltd., Foshan, China) was used to place the spinning liquid mentioned above at an ambient temperature of 35 °C, and the machine parameters were adjusted as follows: the liquid supply speed was 0.4 mL/h, the height was 2 mm, the spinning voltage was 2 kV, the moving speed of the collection plate was 80 mm/s, the grid spacing was set to 1 mm × 1 mm, and the number of layers was 20. The fiber membrane spinning was completed, and, finally, the sample was placed in a drying oven (DHG-9070, electric blast drying oven) at 40 °C for 12 h and saved for later use. The following (Figure 1) is a flowchart of configuring spinning solution and preparing spinning fiber membrane using near-field direct writing electrospinning technology. 

### 2.3. Characterization

#### 2.3.1. Mechanical Property Test

Cut different fiber samples into 10 mm × 50 mm strip samples, record the original gauge length, sample width, and thickness, and obtain the test results through four parallel experiments on a tensile testing machine (AGS-X, Shimadzu Corporation, Tokyo, Japan) [38].

#### 2.3.2. FTIR Analysis

With potassium bromide as the background, the ERY/CSC-I/PCL composite fiber membrane was cut into small pieces with a length of less than 1 mm, and a Fourier transform infrared spectrometer (Nicolet IS10, Thermer Fisher Technology, Waltham, MA, USA) was used to conduct infrared spectrum testing and analysis together with the powdered raw materials to detect composite changes in the functional groups of fibers [39].

#### 2.3.3. XRD Analysis

The sample to be tested was cut into small pieces with a length of less than 1 mm, and an X-ray diffractometer (Ultima IV, Rigaku, Tokyo, Japan) (Cu Kα irradiated radiation) was used to evaluate the crystal phase of the sample. The measurement was carried out in the 2θ range of 10–80°, with a scan rate of 10°/min, using copper target radiation (λ = 1.542) operating at 30 kV and 20 mA.

#### 2.3.4. Thermal Stability Analysis

After cutting the finished product to be tested into small pieces, weigh and record as required, use a thermogravimetric analyzer (TG, NETZSCH Instrument Manufacturing Co., Ltd., Selb, Germany) to perform a thermogravimetric test at a heating rate of 20 °C/min, and perform data processing on the test results. Obtain the TG/DTG diagram.

#### 2.3.5. SEM Analysis

The sample was cut into 6 mm diameter discs, and a vacuum coating instrument (EM ACE600, Leica Instruments Co., Ltd., Wetzlar, Germany) was used to complete the gold plating operation on the sample to be tested. The apparent morphology of the samples was observed and analyzed under a scanning electron microscope (EVO MA 15, Carl Zeiss Optics (China) Co., Ltd., Guangzhou, China).

#### 2.3.6. Hydrophilicity and Water Absorption Performance Test

(1) Hydrophilicity: The hydrophilicity of the material surface can be tested by an optical contact angle measuring instrument (OCA20,DataPhysics, Filderstadt, Germany). The contact angle of the ERY/CSC-I/PCL composite fiber membrane with different gradient precursors and different drug concentrations is measured to evaluate the hydrophilic and hydrophobic properties of the fiber membrane [40].

(2) Water absorption performance test: Cut the finished dressing to be tested into long strips of 10 mm × 50 mm, record its weight, then place it in a test tube containing phosphate buffer solution (PBS, pH = 7.4). Keep at a constant temperature of 37 °C, take out after 24 h, hang above the liquid level for 30 s, and then weigh it. The hygroscopic swelling ratio formula is used for data calculation, and the water absorption performance of the material is analyzed through images.
(1)$$\mathrm{Moisture}\;\mathrm{absorption}\;\mathrm{rate}=\frac{m_{w}-m_{d}}{m_{d}}\times 100\%$$where *m_d_* is the initial dressing mass and *m_w_* is the dressing mass after 24 h of immersion.

#### 2.3.7. Determination of Drug Sustained-Release Performance In Vitro

This experiment uses methyl green fading spectrophotometry. Determine the sustained release of composite fiber membranes with different ERY contents.

(1) Maximum absorption wavelength measurement: Dissolve the methyl green solution and the methyl green reference solution with erythromycin in deionized water to make a solution containing 0.10 mg per 1 mL, and react at 40 °C. 20 min, using a UV–visible spectrophotometer (UV-5100, Shanghai Yuanxi Instrument Co., Ltd., Shanghai, China) in the wavelength range of 500 nm to 800 nm to measure the absorbance. It was found that the maximum absorption was at the wavelength of 626 nm, so 626 nm was selected. Detect wavelength.

(2) Draw the standard curve of erythromycin: Prepare the methyl green working solution as follows: Accurately weigh 0.10 g of the methyl green standard to prepare a 500 mL aqueous solution, shake well, and set aside. Take PBS buffer and prepare an erythromycin stock solution with a concentration of 100 μg/mL. Take a 10 mL centrifuge tube, add 1 mL of methyl green solution and the corresponding erythromycin stock solution, and dilute to a concentration gradient of 5 μg/mL, 10 μg/mL, 20 μg/mL, 30 μg/mL, 40 μg/mL, 60 μg/mL, and 80 μg/mL solutions. React at 40 °C for 20 min, and then measure the absorbance of gradient erythromycin solutions with different concentrations at a wavelength of 626 nm. The working equation obtained through calculation and fitting is A = −0.0005c + 0.1808, R^2^ = 0.9992.

(3) Drug sustained-release test: Weigh a piece of dressing of appropriate size from each of the four groups of finished products with different drug contents, record their mass, and place them in 50 mL of PBS at a constant temperature of 37 °C. After sampling 1 mL with a pipette at time points of 1, 2, 3, 4, 5, 7.5, 10, 12, 24, and 48 h, add 1 mL of methyl green solution, dilute 5 times with PBS, and incubate at 40 °C. React for 20 min and measure the absorption wavelength of the solution at 626 nm, then substitute it into the erythromycin standard curve to obtain the corresponding erythromycin concentration. Compared with the drug concentration when the dressing is fully released, the drug release at a certain moment is obtained. Rate and evaluate the coating effect of the matrix on the drug.
(2)$$\mathrm{Percentage}\;\mathrm{of}\;\mathrm{cumulative}\;\mathrm{drug}\;\mathrm{release}=\frac{w_{t}}{w_{0}}\times 100\%$$
where *w_t_* is the drug release amount at different time points and *w*_0_ is the content of ERY in the composite fiber [41].

#### 2.3.8. Antibacterial Performance Test

In this experiment, *Escherichia coli* (*E. coli*) and *Streptococcus thermophiles* (*S. thermophiles*) were selected to evaluate the antibacterial performance of ERY/CSC-I/PCL composite fiber membranes with different ERY contents. The experimental supplies were pretreated with high-temperature or ultraviolet sterilization. Dilute the activated bacterial solution 100 times on the ultra-clean workbench (SW-CJ-1F, Shanghai Boxun Industry & Commerce Co., Ltd., Shanghai, China) and transfer 100 μL evenly on the surface of the solid culture medium. Place a fiber sample disc with a diameter of 6 mm, seal the culture medium, and place it in a constant-temperature incubator (SPX-150C, Shanghai Boxun Industry & Commerce Co., Ltd., Shanghai, China) at 37 °C for 24 h. Finally, observe the diameter of the antibacterial ring of samples with different drug contents for antibacterial effect evaluation. Repeat the experiment three times and take the average value [42].

## 3. Results and Discussion

In this paper, the CSC-I component in the composite fiber membrane mainly serves as a base material, and we also expect it to replace part of the synthetic polymer PCL to improve the hydrophilicity and humectancy of the material (in the hydrophilicity of the material can be effectively improved with a CSC-I content of 20%). In addition, when the content of CSC-I is high (30%), it tends to clog during the spinning process, so 20% CSC-I was chosen as the follow-up material in the experiment. The content of CSC-I will be 20% thereafter, unless otherwise specified.

### 3.1. SEM Analysis

As shown in Figure 2, from a morphological perspective, the CSC-I/PCL composite fiber membrane was relatively smooth and flat before adding ERY. After the addition of ERY, the surface of the ERY/CSC-I/PCL composite fiber membrane started to become rough, and the spinning uniformity decreased with the increase in ERY content. The overall deterioration was worse, but the diameter change in the ERY/CSC-I/PCL composite fiber membrane was not obvious. This may be because the addition of ERY worsens the leveling properties of the matrix solution, resulting in an uneven surface after the spinning solvent evaporates.

### 3.2. Mechanical Property Test

Figure 3 shows the mechanical properties of composite fiber membranes tested with different substances. Compared with the pure PCL fiber membrane, the tensile strength and elongation at the break of the CSC-I/PCL composite fiber membrane with the addition of CSC-I decreased. After the addition of ERY, with the increase in drug content, the tensile strength of the ERY/CSC-I/PCL composite fiber membrane was worse and the elongation at break was lower, changing from 7.32 (±0.14) MPa to 5.41 (±0.22) MPa. This took place because the incorporation of CSC-I and ERY weakened the entanglement between the chains of PCL molecules, and the effect was more pronounced at higher drug contents.

### 3.3. Hydrophilicity Analysis

As shown in Figure 4, in the CSC-I/PCL composite fiber membrane, as the CSC-I content increases, the contact angle of the material with water gradually decreases from 86.68° to 75.68°. This phenomenon indicates that the addition of CSC-I improves the hydrophilicity of the material. The reason for this is the presence of a large number of hydroxyl structures in the collagen molecule, since their hydrogen bonding with water molecules enhances the hydrophilicity of the material. In the ERY/CSC-I/PCL composite fiber, the addition of ERY can transform the original hydrophilic material into a hydrophobic material, and as the ERY content increases, the contact angle of the material with water gradually increases from 98.40° to 107.7°, indicating that the addition of ERY reduces the material’s hydrophilicity. This happens because ERY molecules contain more hydrophobic aromatic ring structures, and the hydrophobicity is enhanced with the increase in ERY content. In addition, the addition of ERY also increases the surface roughness of the material (as shown in Figure 2), which is also an important factor in the enhancement of hydrophobicity [43].

### 3.4. Water Absorption Performance Test

As shown in Figure 5, the water absorption of all fibrous membranes exceeded 450%. This is mainly due to the fact that the near-field direct-writing electrospinning technique can prepare an ordered and uniform fiber network structure, which has a larger specific surface area and more pores and, thus, improves the water retention effect of the fibrous membranes. The water absorption rate of the composite fiber membrane after adding CSC-I was slightly higher than that of the pure PCL fiber membrane, which might be due to the hydrogen bonding between CSC-I molecules and water molecules. It is worth noting that the water absorption rate of the fibrous membrane also increased after adding ERY. Combined with the change in the surface morphology of the fibrous membrane after adding ERY, it is assumed that this is caused by the special surface structure of the fibrous membrane. From another perspective, the ERY/CSC-I/PCL fibrous membrane showed a more wrinkled surface as a whole, and on the other hand, the dissolution and release of ERY from the fibers could leave tiny spaces, which increased the water absorption rate for the fibrous membrane. These increased the retention of water in the fiber membrane, thus increasing the water absorption rate of the fiber membrane. Especially when the ERY content reached 5%, the fiber membrane showed more wrinkled surfaces, and its water absorption rate was significantly enhanced.

### 3.5. FTIR Analysis

As shown in Figure 6, several obvious PCL characteristic peaks can be found in pure PCL, namely 1176 cm^−1^ (C-O-C asymmetric stretching), 1739 cm^−1^ (carbonyl stretching), 2952 cm^−1^ (CH_2_ asymmetric stretching), and 2867 cm^−1^ (CH_2_ symmetric stretching). CSC-I has a typical amide characteristic absorption peak, and in the CSC-I/PCL composite fiber membrane, compared with pure PCL, there are -NH- stretching vibration peaks (3440 cm^−1^) and amide group stretching vibration peaks (1656 cm^−1^). This is similar to the result of Ghosal et al. [44] and indicates that CSC-I has been successfully loaded onto PCL. The characteristic absorption peaks of ERY are 1721 cm^−1^ (carbonyl stretching), 2884 cm^−1^ (CH_2_ symmetric stretching), and 2975 cm^−1^ (CH_2_ asymmetric stretching) [45]. The above-mentioned characteristic peaks of ERY are consistent with the characteristics of PCL. The peaks overlap each other, but the characteristic peak absorption of the corresponding band is slightly enhanced after ERY is added, from which the successful loading of ERY can be determined.

### 3.6. XRD Analysis

It can be clearly seen in Figure 7 that pure PCL has three typical characteristic diffraction peaks (21.3° (110), 21.9° (111), and 23.6° (200)), which is in agreement with the results of Cai et al. [46]. In the composite fiber membrane with added ERY, no obvious ERY diffraction peaks were found, and there was only a tiny bump near 2θ = 14–16°, which indicated that the content of ERY in the composite fiber membrane might have been too low and the intensity of the diffraction peaks too weak, or ERY might have been distributed in the fibers in an amorphous state without crystallization.

### 3.7. Thermal Stability Analysis

The TG diagram is shown in Figure 8a. The thermal weight loss of the ERY/CSC-I/PCL composite fiber membranes can be divided into three stages, with the first stage showing a small amount of mass loss around 100 °C, which is due to the evaporation of a small amount of water adsorbed on the sample. The second stage is an obvious mass loss near 260–360 °C, which includes thermal degradation of CSC-I and ERY. The third stage is a substantial mass loss near 370–440 °C, which includes the thermal degradation of CSC-I and PCL, which leaves behind ash after the release of gas. Combined with the DTG curves shown in Figure 8b, it can be seen that the maximum weight loss temperatures of both ERY and CSC-I were near 330 °C, and the maximum weight loss temperature of PCL was 407 °C. In the ERY/CSC-I/PCL composite fiber membranes, the maximum weight loss temperatures of the substances did not vary much, and the final mass of the residual ash increased with the increase in ERY content. It is worth noting that compared with pure CSC-I, the onset decomposition temperature of the CSC-I component in the ERY/CSC-I/PCL composite fiber membranes increased from about 200 °C to about 250 °C, which indicates that ERY/CSC-I/PCL composite fiber membranes have better thermal stability than CSC-I.

### 3.8. Determination of Drug Sustained-Release Performance In Vitro

In vitro drug sustained-release experiments are shown in Figure 9. ERY in ERY/CSC-I/PCL composite fiber membranes with different ERY contents was rapidly released in the first 5 h, and then the drug was slowly released for more than 12 h. This suggests that the composite fiber membrane has some sustained-release function. In addition, the different composite fiber membranes showed almost complete release within 24 h, and the release rates were all around 80%. Among them, the overall release rate of the 3% ERY/CSC-I/PCL composite fiber membrane was faster, but the drug release rate was higher, indicating that the 3% ERY/CSC-I/PCL composite fiber membrane had a higher drug utilization rate.

### 3.9. Antibacterial Performance Test

As can be seen from Figure 10, there is no bacterial inhibition zone around the CSC-I/PCL composite fiber membrane, indicating that neither CSC-I nor PCL itself has an antibacterial effect on *E. coli* and *S. thermophilus*. After adding ERY, the average diameters of the ERY/CSC-I/PCL composite fiber membranes with ERY content of 1, 3, and 5% against *E. coli* were 15.17 mm, 18.01 mm, and 18.61 mm, respectively. The antibacterial effect gradually increased as the drug concentration increased. When the ERY content reached 3%, the ERY/CSC-I/PCL composite fiber membrane had a strong antibacterial effect against *E. coli*. For *S. thermophilus*, when the ERY content in the composite fiber membrane was 1, 3, and 5%, the average diameters of the inhibition zones were 16.22 mm, 17.06 mm, and 18.17 mm, respectively. The antibacterial effect gradually increased as the drug concentration increased, but when the ERY content reached 5%, a larger inhibition zone was formed. This shows that ERY has obvious antibacterial effects on both bacteria, and ERY has a stronger inhibitory effect on *E. coli* than *S. thermophilus*.

## 4. Conclusions

In summary, this experiment successfully used near-field direct-writing electrospinning technology to composite the biocompatible polymer PCL with CSC-I material as a drug-loading precursor to prepare a CSC-I/PCL composite fiber membrane and added the drug to prepare an ERY/CSC-I/PCL drug-loaded composite fiber membrane. The hydrophilicity of the CSC-I/PCL composite fiber membrane and the ERY/CSC-I/PCL composite fiber membrane is related to the CSC-I and ERY contents. The addition of CSC-I enhances the hydrophilicity of the composite fiber membrane, which is contrary to the results of ERY. Regardless of whether it is loaded with drugs or not, the composite fiber membrane can present a complete network structure, and because the mesh structure of the composite fiber membrane has more pores, the material’s water retention effect is improved. In addition, although the surface of the drug-loaded composite fiber membrane is relatively rough and the overall spinning uniformity becomes worse, this also causes more wrinkles to appear on the surface of the fiber membrane, and the moisturizing effect is significantly enhanced. Mechanical property tests show that adding CSC-I or ERY will weaken the entanglement between PCL molecular chains, thereby reducing the tensile strength and elongation at the break of the composite fiber membrane. The ERY-loaded CSC-I/PCL composite fiber membrane provides good inhibition of both *E. coli* and *S. thermophilus* while maintaining a slow release for more than 12 h, features that will provide long-lasting anti-infection protection and contribute to good wound healing. Research based on ERY/CSC-I/PCL drug-loaded composite fiber membranes can provide new approaches and theoretical practices for the personalized preparation of micro/nanofiber dressings by near-field direct-writing electrospinning and their application in the field of skin wound healing and repair. This will contribute to the broadening of medical applications.

## Figures and Tables

**Figure 1 polymers-16-01573-f001:**
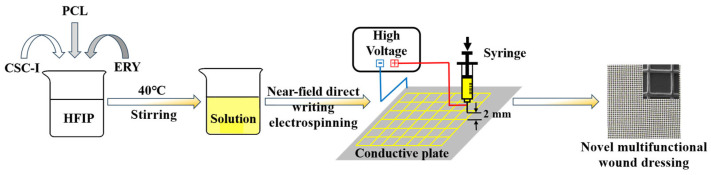
Experimental procedure.

**Figure 2 polymers-16-01573-f002:**
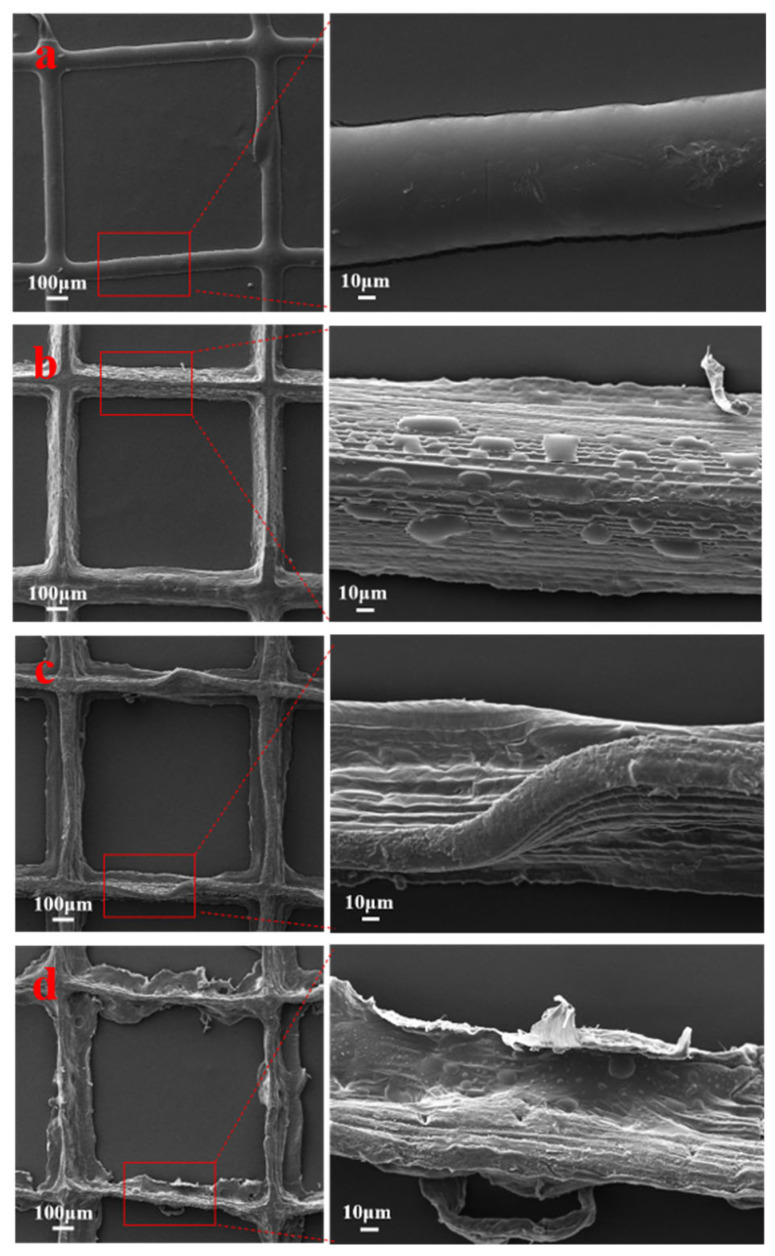
SEM diagrams of composite fiber membranes with different ERY content ((**a**): 0% ERY/CSC-I/PCL; (**b**): 1% ERY/CSC-I/PCL; (**c**): 3% ERY/CSC-I/PCL; (**d**): 5% ERY/CSC-I/PCL).

**Figure 3 polymers-16-01573-f003:**
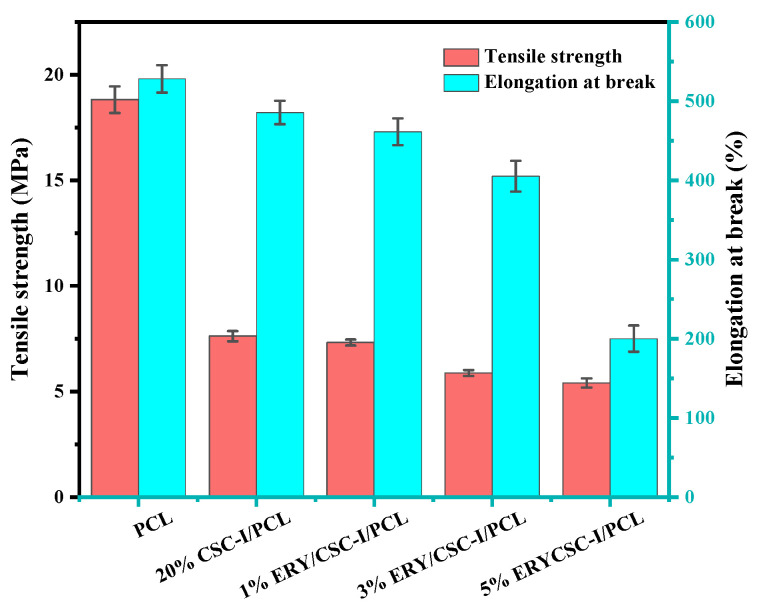
Mechanical property test of composite fiber membranes with different ERY contents.

**Figure 4 polymers-16-01573-f004:**
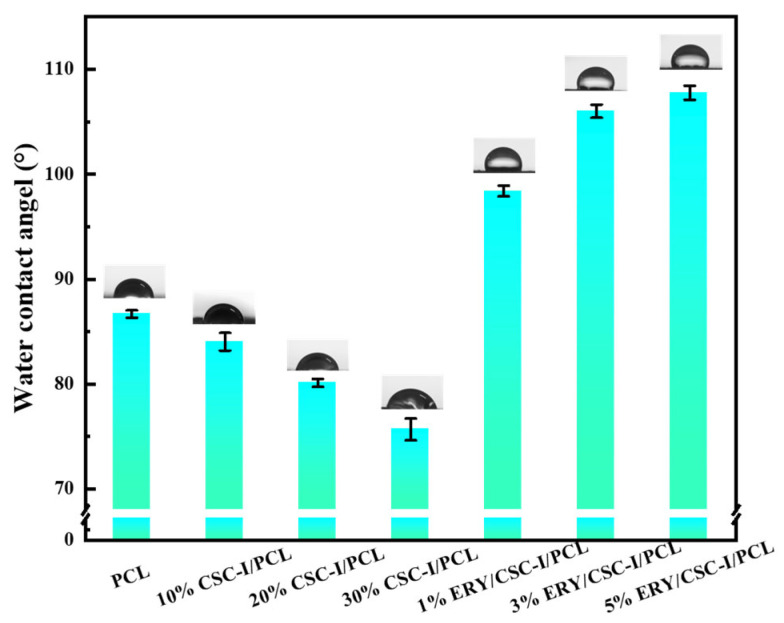
Water contact angle of composite fiber membranes with different CSC-I and ERY contents.

**Figure 5 polymers-16-01573-f005:**
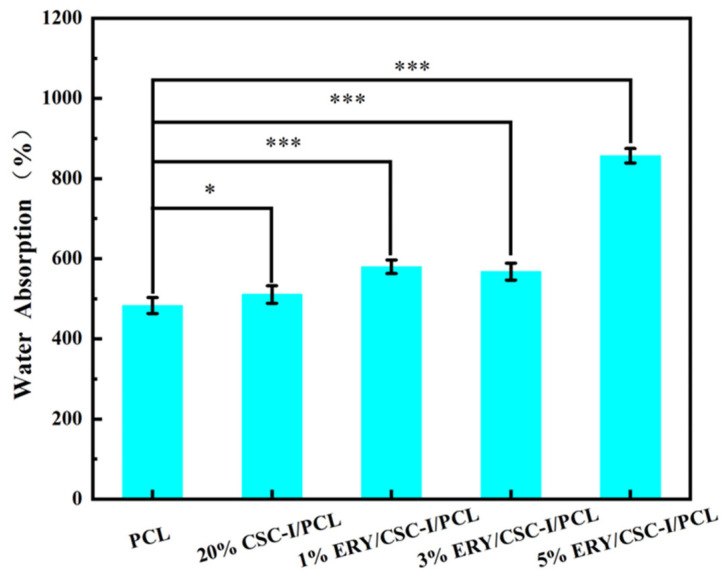
Water absorption rate of composite fiber membranes with different ERY contents (* *p* ≤ 0.05; *** *p* ≤ 0.001).

**Figure 6 polymers-16-01573-f006:**
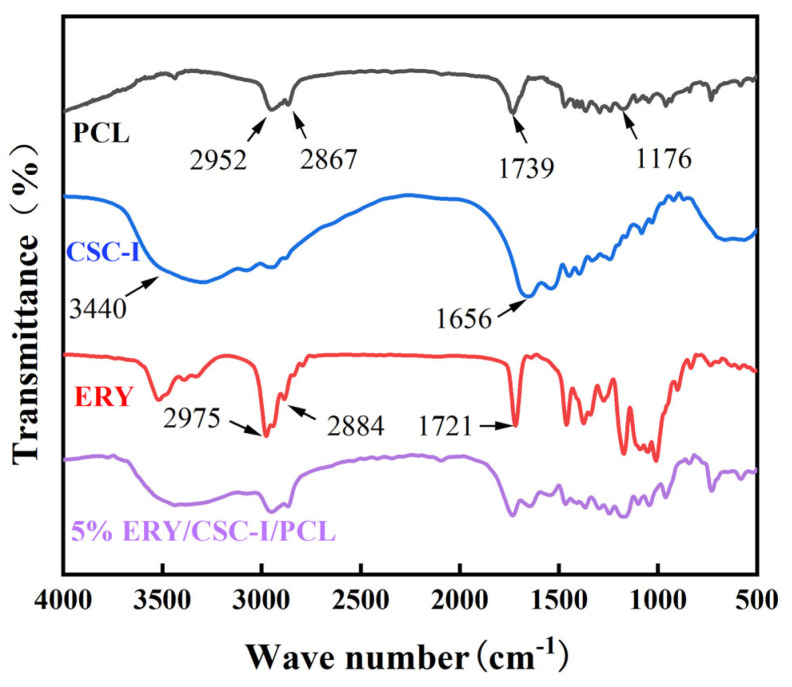
FTIR analysis of composite fiber membranes with different contents.

**Figure 7 polymers-16-01573-f007:**
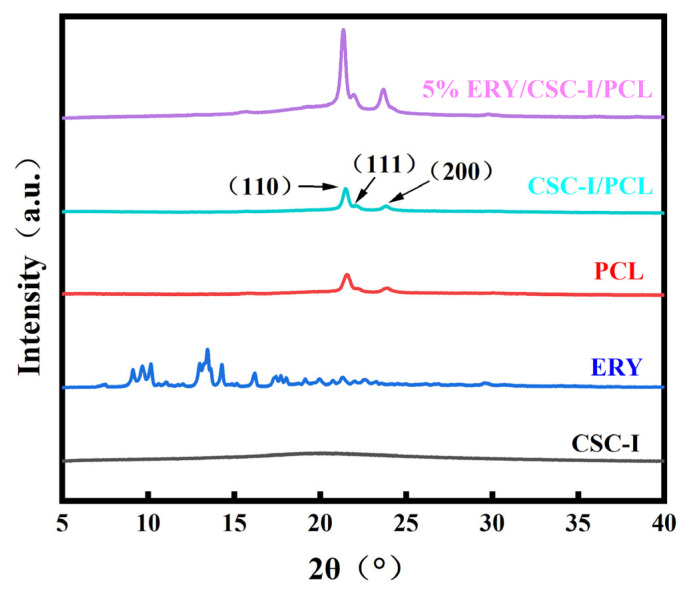
XRD analysis of composite fiber membranes with different contents.

**Figure 8 polymers-16-01573-f008:**
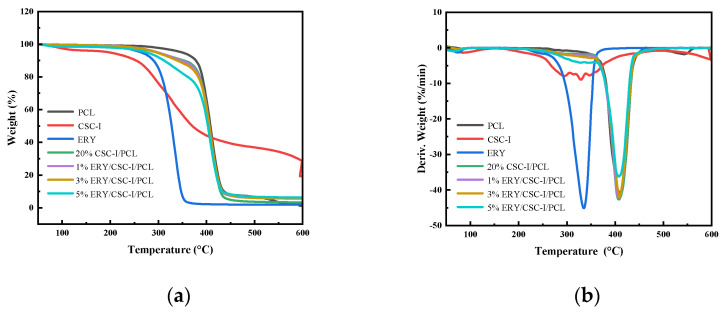
TG (**a**)/DTG (**b**) analysis of composite fiber membranes with different contents.

**Figure 9 polymers-16-01573-f009:**
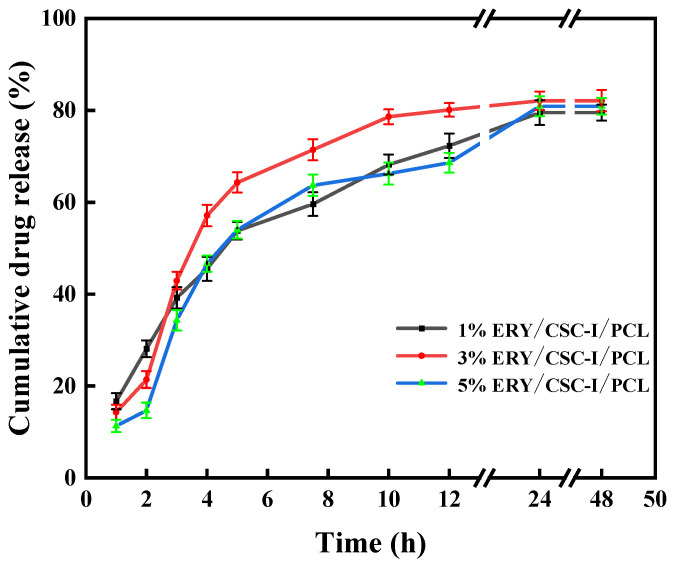
Determination of drug sustained-release performance in vitro.

**Figure 10 polymers-16-01573-f010:**
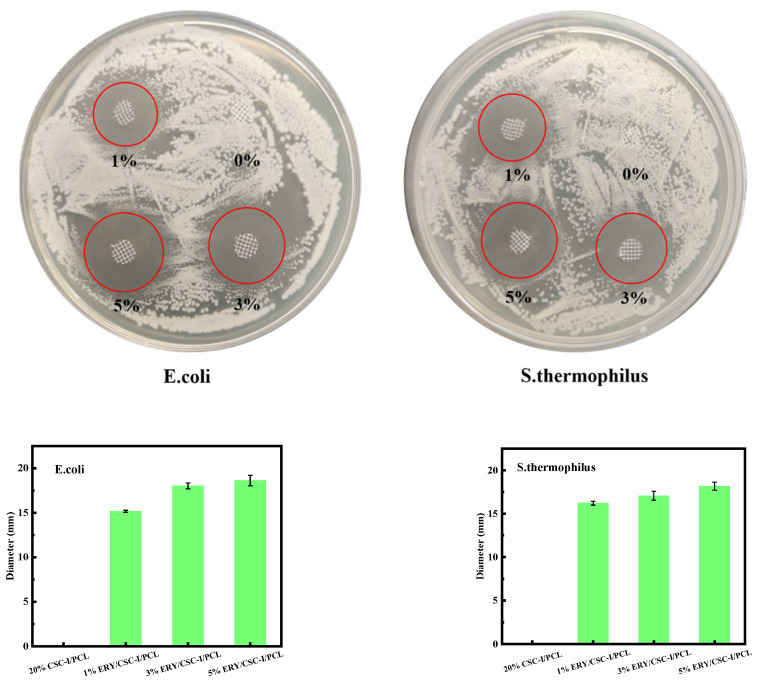
Antibacterial performance test of ERY/CSC-I/PCL samples with different erythromycin contents.

## Data Availability

Data are contained within the article.

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
