# Peer review of "Preparation and Characterization of Novel Multifunctional Wound Dressing by Near-Field Direct-Writing Electrospinning and Its Application"

_polymers, 2024, doi:10.3390/polym16111573_

Round 1

Reviewer 1 Report

Comments and Suggestions for Authors

The article "Preparation and characterization of novel multifunctional wound dressing by near-field direct writing electrospinning and its application" is very interesting and well written. In this article they have investigated the multifunctional wound dressing made of calf skin collagen type I (CSC-I) and polycaprolactone (PCL), hexafluoroisopropanol (HFIP) as solvent and erythromycin (ERY) as anti-infective drug component. The introduction was well written and they reviewed the relevant literature. 

1. Please cite the relevant literature for the methods you have used, especially the methods you have used to characterize?

2. FTIR analysis: Please correct CH2 to CH2 in lines 277, 282, 283.

3. Discussion is missing in this article, please discuss your results in relation to recent relevant articles.

4. Figure 5, To show whether they are significantly different, they need to perform a statistical analysis such as Tukey HSD test or Ttest.

Comments on the Quality of English Language

English language in the article is fine, but only minor editing is required.

Author Response

Dear Editor,

The files including responses, revised manuscripts and author change forms have been uploaded in Zip format files.Thanks.

Best regards,

Wuyi Zhou

Reviewer 2 Report

Comments and Suggestions for Authors

This research introduces a new method for making wound dressings using a technology called near-field direct writing electrospinning. This technique allows them to create precise, mesh-like structures made of tiny fibers. The researchers used a combination of materials (calf skin collagen type I and polycaprolactone) to form the base of the dressing, with a special solvent (Hexafluoroisopropanol) to dissolve them. They also incorporated an antibiotic called erythromycin to fight infection. The products were characterized using SEM, XRD, FTIR and analysed via mechanical, thermal and wettability properties. Finally, antibacterial test was performed. Overall, the manuscript is well written and easy to understand. However, I have the following comments:

 ·       Figure 2, specify if it is SEM for clarify.

·       In section 3.3. Revise the first sentence in the "Hydrophilicity Analysis" section as it seems incomplete and repetitive.

·       All the references are cited in the introduction paragraphs. The results and discussion sections are not detailed and lack citations. Including references would strengthen arguments and acknowledge relevant research.

·       Check the reference list format. Titles ending with "[J]" might be incorrect.

Author Response

(The authors gave the same response as above.)

Reviewer 3 Report

Comments and Suggestions for Authors

In this article, the authors prepared ERY/CSC-I/PCL composites with different content of CSC-I and ERY by near-field direct writing electrospinning. Several characterizations were performed to investigate and compare the properties of the samples. Regarding the data presented in this manuscript, several minor and critical issues require extensive revision from the authors. In this state, I could not recommend publishing this article in Polymers until these issues are properly addressed.

Line 44-45:

“However, they have poor biocompatibility, moisture retention ability, and single function.”

- The authors should add the reference(s) to support this statement.

Line 83-86:

“Calf Skin Collagen Type I (CSC-I ), as a typical collagen, has good hemostatic properties, biocompatibility, low immunogenicity, and biodegradability, it is beneficial to promoting the growth and repair of skin, retains water in the dermis, and maintain its hydration and gloss.”

- Please add the relevant references to support the context line 83-86.

Line 114-115:

“ERY/CSC-I/PCL spinning liquids with different ERY contents were prepared by taking 20% CSC-I and adding 1, 3, and 5% ERY under the same conditions as above.”

- Please provide the reason why the authors choose 20% CSC-I over the other compositions to study the effect of ERY content.

- Also, please provide more details about percentage of ERY content, such as % of PCL mass or something else.

Line 134:

“…membrane was cut into powder form,…”

- It's uncommon that fiber can be cut into powder form. But it can be cut into a specific smaller size or ground into powder. Please add more details on how to prepare the samples for FTIR and XRD measurements.

Line 139-142:

- Please include more details of the XRD measurement parameters such as 2theta range, and x-ray source (Cu Ka, power in kV and mA).

Results and Discussion

- Somewhere in the Results and Discuss section, the authors should briefly discuss why 20 % CSC-I was used instead of 10 and 30 % CSC-I. Currently, only the water contact angle data in Fig.4 shows the results from all CSC-I compositions, but it is still not clear why 20% CSC-I was selected. If these results make the article too long, the authors may include them in the supplementary material.

Line 228:

“Figure 3 shown the mechanical properties…”

- Please correct the underlined word to “showed”.

Line 240-254:

- The discussion line 240-254 requires extensive revision. There are repeated, incomplete sentences that need to be corrected such as line 240-244.

- In addition, the authors should add the reasons why CSC-I and ERY could modify the wetting property of the samples. For example, CSC-I and ERY may contain functional groups that could promote the hydrophilicity or hydrophobicity of the samples. Moreover, there might be other factors that affect the wetting property based on Wenzel and Cassie-Baxter models.

Line 259-270:

- The discussion line 259-270 is difficult to follow. Please properly separate the sentences with appropriate punctuation to discuss the Water Absorption Performance Test.

Line 264-265

“the water absorption rate of the composite fiber membrane after adding CSC-I is significantly higher than that of the pure PCL fiber membrane.”

- Regarding the data shown in Figure 5 of PCL and 20% CSC-I/PCL, the water absorption (%) of both of them seems to line within their error bar ranges, which means they are not statistically different.

- In addition, the authors should show the data measured on 10 and 30% CSC-I and see whether it could support the authors' hypothesis.

Figure 6 and Figure 7:

- Please specify the percentage of ERY in ERY/CSC-I/PCL composite.

Line 294-297:

“In addition, no diffraction peak of ERY is found, which shows that ERY is distributed in an amorphous state in the composite fiber membrane and is not crystalline, which further shows that ERY is well mixed with the base material.”

- Assuming there are 1-5% ERY in CSC-I/PCL samples, these amounts of ERY might be too low to be shown in XRD pattern of ERY/CSC-I/PCL. Also, it is unclear whether the intensities of all data shown in Figure 7 are properly normalized. Therefore, it's hard to believe that ERY could significantly promote the crystallinity enhancement of the sample unless the authors could find the appropriate reference(s) to support this phenomenon.

Line 301-311:

- Extensive revision with proper punctuation is required in this section.

Line 309-310:

“The thermal stability has been improved,…”

- I'm not sure why the authors mentioned that the thermal stability of the samples has been improved at 370-470 C. This is because when other molecules decompose, it means that the sample's mass will eventually decrease to a certain point. More importantly, the authors should discuss each point by referring to Figure 8 (a) and (b), which do not exist in this section.

Line 316-321:

“In vitro drug sustained release experiments showed that ERY in ERY/CSC-I/PCL composite fiber membranes with different ERY contents were released rapidly in the first 5 hours, and then the drug was slowly released for more than 12 hours, indicating that the composite fiber membrane has certain Sustained release function, in addition, different composite fiber membranes are basically completely released within 24 hours, and the release rates are all around 80%.”

- Please revise this long sentence by dividing it into a few more sentences. Also, how do the authors know that ERY in the fiber membranes was completely released within 24 hours? To verify this, the authors should extend the time axis of Figure 9 to over 24 hrs.

Line 350-356:

“Regardless of whether it is loaded with drugs or not, the composite fiber membrane can present a complete network structure, and because the mesh structure of the composite fiber membrane has more pores, the material's water retention effect is improved, on the other hand, although the drug-loaded composite The surface of the fiber membrane is relatively rough, and the overall spinning uniformity becomes worse.”

- Please revise this context with appropriate punctuation.

Author Response

(The authors gave the same response as above.)

Round 2

Reviewer 2 Report

Comments and Suggestions for Authors

As a result of the complete revision undertaken by the authors to incorporate all reviewer feedback, the manuscript now demonstrates exceptional suitability for publication. The proposed revisions successfully support the research and improve the clarity and scientific accuracy of the paper.

Author Response

Dear Editor,

Thank you very much for your kind attention and careful review for our manuscript entitled “Preparation and characterization of novel multifunctional wound dressing by near-field direct writing electrospinning and its application”. We have carefully checked and revised our manuscript according to the suggestions of the reviewers. The responses were shown as follows.

Best regards,

Wuyi Zhou 

Reviewer 3 Report

Comments and Suggestions for Authors

In this version of the manuscript, the authors have addressed most of the comments and questions mentioned in the previous version. However, there are some additional issues that should be corrected before publishing this article.

Line 139-140:

“…composite fiber membrane was cut into small pieces…”

- It would be better to provide the approximate sizes of the fiber cut for these measurements including FTIR and XRD.

Line 143:

“composite Changes in functional groups of fibers”

- changes

Line 252-257:

“As shown in Figure 4, in CSC-I/PCL composite fiber membrane, as the CSC-I content increases, the contact angle of the material to water gradually decreases from 86.68° to 75.68°, indicating that the addition of CSC-I improves the hydrophilicity of the material, the reason for this is the presence of a large number of hydroxyl structures in the collagen molecule, and their hydrogen bonding with water molecules enhances the hydrophilicity of the material. In the ERY/CSC-I/PCL composite fiber.”

- The first sentence in this paragraph is too long and should be properly separated into a couple of shorter sentences.

- Please also correct an incomplete phrase "In the ERY/CSC-I/PCL composite fiber." (line 257) with appropriate punctuation.

Line 260-261:

“…the addition of ERY re-duces the material's Hydrophilic.”

- Please change the underlined words to “reduces” and “hydrophilicity”.

Line 279-282:

“On the one hand, the ERY/CSC-I/PCL fibrous membrane showed a more wrinkled surface as a whole, and on the other hand, the dissolution and release of ERY from the fibers could leave tiny spaces, which increased the water absorption rate for the fibrous membrane.

- Please revise or change “on the other hand” (line 280) to different words.

Author Response

(The authors gave the same response as above.)
